# Invasion of a Horticultural Plant into Forests: *Lamium galeobdolon argentatum* Affects Native Above-Ground Vegetation and Soil Properties

**DOI:** 10.3390/plants12071527

**Published:** 2023-03-31

**Authors:** Hans-Peter Rusterholz, Katharina Huber, Bruno Baur

**Affiliations:** 1Department of Environmental Sciences, University of Basel, Bernoullistrasse 30, 4056 Basel, Switzerland; hans-peter.rusterholz@unibas.ch (H.-P.R.); huber.ka@aol.com (K.H.); 2Program Man-Society-Environment, Department of Environmental Sciences, University of Basel, Vesalgasse 1, 4051 Basel, Switzerland

**Keywords:** biodiversity, clonal regeneration, Ecoplates, *Lamium* sp., native cultivar, ornamental plant, soil enzyme, soil microbial community, stoloniferous plant

## Abstract

Horticultural trade is considered the most important pathway for the introduction of non-native plant species. Numerous horticultural plants are spreading from private gardens and public green space into natural habitats and have the potential to alter native biodiversity and ecosystem functioning. We assessed the invasiveness of the horticultural plant *Lamium galeobdolon* subsp. *argentatum*. We documented its spread in semi-natural habitats in the surroundings of Basel, Switzerland, over the past decades. We compared the performance of *L. g. argentatum* with that of the native subspecies *Lamium galeobdolon galeobdolon* based on surveys in forests and a pot experiment under standardized conditions. We also assessed whether the two subspecies differentially affect native forest vegetation and various physical, chemical and biological soil properties. The horticultural *L. g. argentatum* has tripled its occurrence in forests in the region of Basel in the last four decades. *Lamium g. argentatum* had both a higher growth rate and regeneration capacity than the native subspecies. Furthermore, *L. g. argentatum* reduced native plant species richness and changed the species composition of the ground vegetation, in addition to altering several soil properties in deciduous forests. *Lamium g. argentatum* should therefore be considered an invasive taxon.

## 1. Introduction

Numerous plant species are transported around the world for commercial and horticultural purposes [1,2]. As a side effect, the global horticultural trade became the strongest driver for the introduction of non-native plants [3]. This is because a significant proportion of horticultural plants become naturalized and some even become invasive [4,5]. In Europe, for example, more than 50% of the naturalized non-native plant species have been imported for horticultural purposes [6,7].

Invasive non-native plant species are considered a major threat to native biodiversity [8,9]. They can affect ecosystems by changing native plant diversity and species composition, and in some cases lead to the extinction of native species [10]. Non-native plant species also have the potential to change soil properties including pH and nutrient levels (e.g., phosphorus, nitrogen), and to alter the abundance and composition of soil microbial organisms [11,12]. Furthermore, non-native plants can disrupt symbiotic associations between soil fungi and host plants, which in turn affects biodiversity [13,14,15].

Successful plant invaders are often characterized by a rapid growth rate (including clonal expansion), large seed production, rapid seedling emergence, or high vegetative regeneration capacity after disturbance [16,17]. Any one of these traits, or a combination of them, allows rapid colonization of different types of natural habitats [18]. A comparison of these traits, which are often summarized as performance, with the corresponding traits of closely-related native species allows an assessment of the invasiveness of the non-native species [6,19].

In our study, we compared the performance of the horticultural non-native *Lamium galeobdolon* subsp. *argentatum* (Variegated yellow archangel; Figure 1) escaped in forests with that of the closely-related native *Lamium galeobdolon* subsp. *galeobdolon* (L.) (Yellow archangel; Figure 1) found in the same habitat. The horticultural non-native *L. g. argentatum* (Smejkal) J. Duvign (synonyms: *Galeobdolon argentatum* Smejkal, *Lamiastrum argentatum* (Smejkal) H. Melzer, *Lamium montanum* var. *florentinum* (Silva Tar.) Buttler & Schippmann, *Lamiastrum galeobdolon* subsp. *argentatum* (Smejkal) Stace, *Lamium argentatum* (Smejkal) Henker ex G. H. Loos, *Galeobdolon luteum* var. *florentinum* Silva Tar.) is a stolonifereous herb belonging to the family Lamiaceae. *Lamium g. argentatum* forms stolons up to 1 m long, consisting of multiple nodes with opposite unexpanded leaves and primordial roots at each node, connected by incompletely expanded internodes. The leaves are silvery-colored and mostly with purple-colored spots. The yellow-colored flowers are arranged in 2–10 whorls in a terminal, spike-like inflorescence. The flowers are hermaphroditic and chasmogamous, pollinated by bees and bumblebees. Seeds are dispersed by ants. *Lamium g. argentatum* often forms a dense vegetation layer (Figure 2). *Lamium g. argentatum* has the potential to hybridize with *L. g. galeobdolon* [20], and with *Lamium galeobdolon montanum* ((Pers.) Hayek) [21], which also occurs at low abundance in the study region.

It has been claimed that *L. g. argentatum* is a native cultivar, possibly descended from a variety of *Lamium galeobdolon montanum* (Pers.) Hayek with silver-spotted leaves and large yellow flowers [21,22]. Therefore, the horticultural breed *L. g. argentatum* has no natural distribution area [23]. Plants of this horticultural subspecies spread from gardens to natural habitats in many countries in Europe, North and South America, in Japan, Australia, and New Zealand [24]. Between 1990 and 2020, the number of *L. g. argentatum* records in the wild increased from about 2000 to over 27,000 in Europe [24]. *Lamium g. argentatum* is listed in the Global Register of Introduced and Invasive Species in Germany, Belgium, France, Italy, Czech Republic and Slovenia [24], and is considered an invasive taxon in Sweden [25], Great Britain [26], and in the United States of America [27].

The closely-related native *L. g. galeobdolon* (Figure 1) has the same growth form and very similar plant traits as the horticultural *L. g. argentatum*, but the leaves are green in color and the yellow flowers are slightly smaller [22]. Its distribution area includes Central Europe, Great Britain and Ireland, western parts of Russia, and small areas in Turkey and Northern Iran [28]. Both subspecies are often found together in urban and suburban nutrient-rich, moist deciduous and coniferous forests [23]. In some European countries, *L. g. argentatum* is now more common in forests than the native *L. g. galeobdolon* [23].

There is a gap in knowledge about the effects of the horticultural *L. g. argentatum* on native vegetation and soil properties. To assess the invasiveness of *L. g. argentatum*, we conducted a series of comparative field studies and a pot experiment under standardized conditions, and documented the spread of the subspecies in semi-natural habitats in the surroundings of Basel, Switzerland, in the past decades. In particular, we examined the performance of *L. g. argentatum* and *L. g. galeobdolon* in 12 forests by comparing their patch sizes, plant cover, the number of both stolons and inflorescences, the number of nodes per stolon and stolon length. We also assessed whether the two subspecies have different effects on native forest vegetation as well as on various physical, chemical and biological soil properties. Furthermore, we investigated whether distinct patches of the two subspecies differ in their growth over 7 years. Using stolon fragments, we conducted a pot experiment under standardized conditions to examine whether differences in local environmental conditions could explain the observed differences in performance between the two subspecies.

In our study, we tested the following hypotheses:(1)Illegal garden waste dumping is a major source of non-native plants in suburban and urban forests [29,30], with an increasing frequency in the past decades [31]. We therefore expected that the number of *L. g. argentatum* records at the landscape level has also increased in recent decades.(2)Nehring et al. [23] reported that *L. g. argentatum* is now more common in suburban and urban forests than the native subspecies. This could be due to better performance of the horticultural *L. g. argentatum* compared to the native subspecies. We therefore expected that *L. g. argentatum* has a higher establishment rate of juvenile plants derived from clonal propagation, a greater number of stolons and longer stolons than native *L. g. galeobdolon.* Furthermore, we hypothesized that the horticultural subspecies has a greater patch growth rate and forms larger plant patches than the native subspecies.(3)We also hypothesized that compared to the native *L. g. galeobdolon*, the non-native *L. g. argentatum* reduces the richness of native plants in the ground vegetation, changes the composition of native plant species and soil properties, and alters the abundance and composition of soil microbial organisms.

## 2. Results

### 2.1. Spread of L. g. argentatum in the Region of Basel

The number of mapping areas in which *L. g. argentatum* was found tripled between the two surveys (Figure 3a,b). In 1980–1996, individuals of *L. g. argentatum* were recorded in 18 mapping areas, in 2010–2020 in 54 out of 92 mapping areas (Figure 3a,b). In addition, the frequency of *L. g. argentatum* occurrences within mapping area increased between the two surveys (*Chi*^2^ = 36.5, df = 3, *p* < 0.0001; Figure 4). In the 2010–2020 survey, a higher proportion of mapping areas with two or more records of *L. g. argentatum* was found (Figure 4).

### 2.2. Performance of Lamium subsp. in Swiss Forests

Field data collected in 12 different forests showed that the horticultural *L. g. argentatum* and the native *L. g. galeobdolon* differed in patch size (*F*_1,11_ = 30.84, *p* < 0.001), number of stolons (*F*_1,11_ = 12.34, *p* = 0.005), and stolon length per plant (*F*_1,11_ = 7.54, *p* = 0.019; Table 1). Overall, patches of *L. g. argentatum* were almost ten times larger than those of the native *L. g. galeobdolon* (mean ± s.e: 67.9 ± 15.7 m^2^ vs. 7.1 ± 1.1 m^2^). The horticultural *L. g. argentatum* also had a greater number of stolons (3.4 ± 0.10 vs. 2.7 ± 0.09) and longer stolons (69.8 ± 2.1 cm vs. 63.5 ± 1.9) than the native *L. g. galeobdolon* (Table 1). None of the remaining plant traits measured differed between the two subspecies (in all traits, *p* > 0.24). Interestingly, there was no site effect for any of the measured plant traits (in all cases, *p* > 0.18; Table 1).

### 2.3. Growth of L. g. argentatum and L. g. galeobdolon Patches over 7 Years

Patches of the two subspecies differed in growth over 7 years (*F*_1,6_ = 26.16, *p* = 0.002). Patches of *L. g. argentatum* increased from 1.5 m^2^ to 4.6 m^2^ (mean area covered), while those of *L. g. galeobdolon* increased from 1.2 m^2^ to 1.8 m^2^ (Figure 5). The patch increase per year was 0.51 m^2^ for *L. g. argentatum* and 0.10 m^2^ for *L. g. galeobdolon.*

### 2.4. Performance under Standardized Conditions

The establishment rate of juvenile plants (percentage of juvenile plants that emerged from original stolon fragments per month) in the experimental pots was 41% higher in the horticultural *L. g. argentatum* than in the native *L. g. galeobdolon* (Figure 6; *F*_1,92_ = 53.28, *p* < 0.0001), and was affected by the forest of origin of the stolons (*F*_11,92_ = 53.28, *p* = 0.0410). The establishment rate of juvenile plants increased with the time elapsed in the experiment (*F*_5,590_ = 334.55, *p* < 0.0001; Appendix A). In addition, there was an interaction between subspecies and origin (Figure 6; *F*_11,92_ = 5.75, *p* < 0.0001), as the establishment rate of *L. g. argentatum* was higher than that of *L. g. galeobdolon* in eight forests of origin, equal in three forests of origin, and even lower in one forest of origin. Finally, differences in the establishment rate of juvenile plants of the two subspecies between March and April caused the significant interaction between subspecies and time (Figure 6; *F*_5,590_ = 9.47, *p* < 0.0001).

Stolons emerging from juvenile plants were only recorded between June and July towards the end of the experiment (Appendix A). The two subspecies did not differ in stolon emergence (percentage of stolons that emerged from juvenile plants; *F*_1,92_ = 0.12, *p* = 0.722). However, the percentage of stolons that emerged from juvenile plants was affected by node position in the original stolon (*F*_4,92_ = 2.50, *p* = 0.048), and the forest of origin (*F*_11,92_ = 2.74, *p* = 0.004). In both subspecies, the percentage of stolons that emerged from juvenile plants was highest in fragments originated from the third node (17.5%), and lowest in fragments from the first node (9.2%). The percentage of emerged stolons of both subspecies combined varied from 6% (Zurich) to 26% (Seprais). The significant interaction between percentage of emerged stolons and forest of origin resulted from a lower percentage of emerged stolons in *L. g. argentatum* in three forests than that of *L. g. galeobdolon*.

The biomass of juvenile *L. g. argentatum* was 12% higher than that of *L. g. galeobdolon* (Appendix A; *F*_1,91_ = 4.90, *p* = 0.029). Furthermore, the biomass of juvenile plants was influenced by node position (*F*_4,91_ = 5.43, *p* < 0.001), and forest of origin (*F*_11,91_ = 3.63, *p* < 0.001). In both subspecies, the biomass of juvenile plants originated from the first node was 40% lower than that from the other four nodes. In addition, there was an interaction between subspecies and forest of origin (*F*_11,91_ = 3.79, *p* < 0.001), as the biomass of juvenile *L. g. argentatum* was larger in nine forests of origin than that of *L. g. galeobdolon*, while the opposite was found in three forests of origin.

### 2.5. Impact on Native Ground Vegetation and Soil Properties

#### 2.5.1. Native Ground Vegetation

Of a total of 61 plant species recorded in the ground vegetation, 34 species (55.7%) were found in *L. g. argentatum* plots, 49 (80.3%) in *L. g. galeobdolon* plots, 40 (65.7%) in *L. g. argentatum* control plots, and 38 (62.2%) in *L. g. galeobdolon* control plots (Appendix A). Plant species richness (number of species per 3 m^2^) was significantly reduced in *L. g. argentatum* plots compared to that in *L. g. galeobdolon* plots (Figure 7, Table 2). This reduction in species richness was more pronounced in spring (62%) than in autumn (45%). The significant interaction between subspecies and season was due to the seasonal differences in plant species richness in plots of the two subspecies (Figure 7, Table 2). Plant species richness was also positively related to the ground vegetation cover (those of the *Lamium* subspecies excluded) in all plot types in spring and autumn (in all cases *p* < 0.001).

Analysis of similarity (ANOSIM) revealed that species composition of the ground vegetation differed significantly between spring and autumn (*R* = 0.504, *p* < 0.001), and between the three forests (*R* = 0.609, *p* < 0.001). Overall, the two subspecies caused different shifts in plant species composition in each of the three forests (Allschwil: *R* = 0.219, *p* = 0.001; Bottmingen: *R* = 0.080, *p* = 0.017; Riehen: *R* = 0.137, *p* = 0.001).

#### 2.5.2. Physical and Chemical Soil Properties

Plant-available phosphorus content was higher in *L. g. argentatum* plots than in *L. g. galeobdolon* plots (Figure 8, Table 2). There was a tendency for a slightly larger difference in plant-available phosphorus content between plots of the two subspecies in spring than in autumn (Figure 8, Table 2). Soil moisture was not affected by the two subspecies (Table 2). However, the significant interaction between subspecies and season was the result of a slightly higher soil moisture in *L. g. galeobdolon* plots than in *L. g. argentatum* plots in spring, while the opposite was found in autumn.

#### 2.5.3. Soil Enzyme Activity

Acid phosphatase activity (nmol MUB released per g soil and h) was significantly lower in spring than in autumn (Figure 9). In spring, acid phosphatase activity did not differ between plots with either *L. g. argentatum* or *L. g. galeobdolon* and control plots (Figure 9). In autumn, however, acid phosphatase activity was higher in *L. g. argentatum* plots than in *L. g. galeobdolon* plots (Figure 9, Table 2).

#### 2.5.4. Soil Fungal and Bacterial Community Profiles

The number of bacterial operational taxonomic units (OTUs) was significantly lower in *L. g. argentatum* plots than in *L. g. galeobdolon* plots (Figure 10, Table 2). The significant interaction between subspecies and forest was due to a differential reduction in the number of bacterial OTUs in *L. g. argentatum* plots in the three forests (Table 2). The significant interaction between subspecies, forest and season indicates a high seasonal variation in the number of bacterial OTUs across different plot types and forests (Table 2).

#### 2.5.5. Physiological Profiles of Soil Bacteria

Metabolic activity of the soil microbial community indicated by the average well color development (AWCD) was not influenced by the two *Lamium* subspecies (Table 2). ANOSIM analysis based on AWCD data also showed that the CLPP patterns did not differ between *L. g. argentatum* and *L. g. galeobdolon* plots in spring or autumn in any forest (in all cases, *p* > 0.101). In contrast, substrate richness was significantly higher in *L. g. argentatum* plots than in *L. g. galeobdolon* plots in both spring and autumn (Figure 11, Table 2).

## 3. Discussion

### 3.1. Spread of L. g. argentatum in the Region of Basel

During recent decades, the trade in horticultural plants has increased as more geographic regions have been explored as new sources for horticultural plants [33,34]. The spread of horticultural plants (e.g., *Sedum stoloniferum*, *Cotoneaster horizontalis*) from private gardens and/or public green spaces in semi-natural habitats is well documented [29,30,35]. The three-fold increase in the occurrence of *L. g. argentatum* recorded in our study in the Basel region between the two surveys is similar to those found in the southern Swedish province of Scania between 1980 and 2020 (250%; [36]) and in Britain and Ireland in the same period (260%; [37]). The large increase in the occurrence of *L. g. argentatum* in the wild could be the result of several factors. In the 1960s to 1980s, *L. g. argentatum* was a popular horticultural species for decorating private gardens and public green spaces [38]. After that, *L. g. argentatum* has been replaced by new, more attractive horticultural plant species. As a result, *L. g. argentatum* was often dumped together with other garden waste in semi-natural and natural habitats [31]. Illegal dumping of garden waste into forests adjacent to settlement areas are known sources of the spread of non-native plant species [29,30,31]. However, the establishment and growth of new populations take time, and this can result in delayed awareness of the presence of new plant species in a given habitat. In the case of *L. g. argentatum*, changes in taxonomy and difficulties in distinguishing this subspecies from different native *Lamium* subspecies could lead to misidentification of the ornamental plant, and thus further delay the awareness of its spread. For example, *L. g. argentatum* was considered as a native subspecies of *Lamium galeodolon* in the past [22].

*Lamium g. argentatum* has the potential to hybridize with the native *L. g. galeobdolon* [20]. Wegmüller [39] reported that the hybrid *L. g. galeobdolon* × *L. g. flavidum* produced few seeds that did not germinate and crosses between *L. g. galeobdolon* × *L. g. montanum* were not successful. So far, however, there is no study that has quantitatively examined this aspect.

### 3.2. Performance of Lamium subsp. in Swiss Forests

Assessing whether certain plant traits determine the invasiveness of non-native plant species is a prerequisite both for predicting which non-native species will become invasive and for managing already established non-native species [40]. One approach used to achieve this goal is to compare traits of non-native species with native species in a given environment [6,16]. In our study, the non-native *L. g. argentatum* had greater numbers and longer stolons, and a higher growth rate than the native *L. g. galeobdolon*. These differences in performance may account for the proven successful invasion of the non-native *L. g. argentatum* in the wild. A pilot study showed that apart from the difference in the number of inflorescences, the two subspecies did not differ in seed set, seed weight and germination rate [41]. This finding rules out the possibility that differences in reproductive performance could be another factor affecting the invasion success of *L. g. argentatum.* Our finding that growth rate is an important factor in the invasiveness of the non-native horticultural *L. g. argentatum* coincides with the results of a global meta-analysis [16]. Furthermore, the similarity of the non-native plant species to the native one, as in the present study, is assumed to be another factor favoring the invasiveness of non-native species [42].

### 3.3. Performance under Standardized Conditions

Many clonal invasive plants form dense monocultures by outcompeting native species, which can alter the structure and function of invaded ecosystems [43]. Clonal growth includes physiological integration and clonal storage organs [44,45]. Stolon storage resources become available for clonal fragment regeneration [45]. The higher regeneration capacity (expressed as a percentage of successfully established juvenile plants) of the non-native *L. g. argentatum* than the native *L. g. galeobdolon* found in our study agrees with the results of He et al. [46], who showed that a high regeneration capacity of clonal fragments can contribute to the invasiveness of non-native plants. In contrast, Song et al. [47] reported that regeneration capacity of clonal fragments did not differ between the invasive plant *Alternanthera philoxeroides* and its congeneric native *Alternanthera sessilis*. Furthermore, the regeneration capacity of 14 introduced stoloniferous species was not higher than that of 25 native stoloniferous species in China [48]. As reported in other studies, the higher biomass of the established juvenile plants of *L. g. argentatum* found in our study could be related to the higher regeneration capacity of the non-native plants [46,47]. Thus, the effect of clonal fragment regeneration on invasiveness of non-native plants is variable and seems to depend on the species. However, the regeneration capacity of clonal fragments is important for population dynamics and can contribute to the spread and colonization of non-native clonal plants [49].

### 3.4. Impact on Native Ground Vegetation and Soil Properties

In forests, non-native plants have the potential to alter vegetation structure by reducing the richness of native plant species and altering plant species composition [30,50,51]. The extent of reduction in plant species richness in ground vegetation recorded in our study (45–62%) is in the range observed for other non-woody invasive species including *Lupinus polyphyllus* (21% [52]), *Solidago canadensis* (66%; [53]), *Reynoutria japonica* (50% [54]), and for woody invasive species such as *Robinia pseudoacacia* (42–54% [55]), *Lonicera maackii* (50% [56]), and *Prunus laurocerasus* (40% [30]). Furthermore, the *L. g. argentatum*-induced changes in species composition parallel the results of several studies showing that non-native plants have the potential to alter native species composition [30,51,57]. In our study, the change in plant species composition was more pronounced in spring than in autumn. This can be explained by the general reduction in both the abundance and richness of vernal geophytes in *L. g. argentatum* plots (e.g., *Anemone nemorosa*, *Ranucultus ficaria*, *Arum maculatum*). Here, we have to mention that the native *L. g. galeobodolon* did not affect the richness or composition of native plant species.

Plant-induced changes in physical, chemical and biochemical soil properties, as well as the activity and composition of microbial communities, can play a key role in the invasion success of non-native plant species [58,59,60]. However, a wide range of impacts on non-native plants on given soil properties have been identified [12]. Our finding that soil moisture was not affected by *L. g. argentatum* contrasts with the results recorded in other invasive plants [15,61,62]. Furthermore, we found that the presence of *L. g. argentatum* increased plant-available phosphorus in the soil. This increase could be due to the high activity of acid phosphatase and the substrate richness, both of which can increase the mineralization rate of phosphorus in the soil. Another possibility could be that the two subspecies colonize forest sites with different levels of phosphorus in the soil. However, our small-scale study design and similar plant available phosphorus levels in the control plots and *L. g. galeobdolon* plots exclude this possibility. Similar changes in plant-available phosphorus have been reported for *Ambrosia artemisiifolia* [63] and *Ageratina adenophora* [64]. In contrast, the invasion of *Solidago canadenis* [53] and *Chromolaena oderata* [64] did not affect plant-available phosphorus.

Soil enzymes mediate and catalyze a number of biochemical processes in soil that are essential for the provisioning and recycling of soil nutrients [65]. The high activity of phosphatase and that of the inter-correlated β-glucosidase in soil invaded by *L. g. argentatum* is in line with the results of a global meta-analysis showing that invasive plants increase these soil enzymes [66]. It is therefore assumed that the higher activity of these soil enzymes after plant invasion is due to invasive plants with high nutrient requirements competing with soil microbial communities for available nutrients [67], and that effective limitation might stimulate enzyme production [68].

The soil microbial community is an essential component of the soil ecosystem and influences processes such as soil formation and fertility, nutrient turnover and carbon storage [69]. In our study, the presence of *L. g. argentatum* reduced soil bacterial richness (number of bacterial OTUs). Similarly, the invasive plant *Rhus typhina* reduced soil bacterial richness [70]. Other studies, however, reported no effect of invasive plants on soil bacterial richness (e.g., *Solidago canadensis* [71], *Alliaria petiolata* [72]). A meta-analysis revealed that besides a high variation in the impact of invasive plant species on soil bacterial communities, only allelopathic invasive plants increased soil bacterial diversity [73]. Soil bacterial richness was also positively affected by the magnitude of cover of the invasive plant *Bromus inermis* [74]. In addition, under conditions of reduced native plant species diversity, soil microbial diversity was lower [75,76]. Thus, the reduced bacterial richness recorded in our study could be due to the reduced plant species richness recorded in *L. g. argentatum* plots. However, this was not the case in our study because no relationship between the number of bacterial OTUs and plant species richness was found. The unexpected result could be a result of contrasting effects of changed soil properties and of both altered plant species richness and composition on soil bacterial richness.

The carbon utilization pattern (AWCD) represents a suitable indicator for the overall metabolic activity of soil microbial communities [63,77]. In our study, the presence of *L. g. argentatum* did not change the activity of the carbon sources. A similar lack of impact was noted in other invasive plants including *Reynoutria japonica* [78] and *Bidens pilosa* [79]. In contrast, various other studies demonstrated that non-native plants increase the metabolic activity of soil microbial organisms [30,63,80]. We also recorded an increase in substrate richness in soils invaded by *L. g. argentatum*. This result is supported by various other studies [63,80,81].

In our study, the presence of *L. g. argentatum* did not alter the carbon utilization pattern. This indicates that *L. g. argentatum* had no effect on the catabolic potential and functional diversity of the soil microbial community. In general, the different impacts on soil properties could be caused by differences in plant traits of the non-native species involved [11], differences in habitat and soil types [53], and by the extent of the invasion and its history [82].

## 4. Materials and Methods

### 4.1. Spread of L. g. argentatum in the Region of Basel

Brodtbeck et al. [32] provided detailed information on the occurrence and frequency of *L. g. argentatum* in the surroundings of Basel, Switzerland, in the years 1980–1996. Their study area covered approx. 765 km^2^, including border areas of France (Alsace) and Germany, divided into 92 units (mapping areas) ranging in size from 3 to 20 km^2^ [32]. For each mapping area, the frequency of *L. g. argentatum* occurrence was presented using the following classes: 0, no record; 1, a single record; 2, 2–3 records; and 3, 4 and more records (Figure 3a). Using the same methods, we repeated the survey in the years 2010–2020 (Figure 3b).

### 4.2. Performance of Lamium subsp. in Swiss Forests

We used the distribution map of *L. g. argentatum* [83] to select 12 forests with occurrences of this subspecies (Table 1). Eleven of the forests were evenly distributed over the Swiss Plateau and one forest was in Germany near the Swiss border (Table 1). In each of the 12 forests, we chose a study area of 50 m × 100 m containing both patches of *L. g. argentatum* and patches of native *L. g. galeobdolon*. We randomly selected six patches (>1 m^2^) from both taxa. We measured the area of each patch (to the nearest 0.5 m^2^) and visually estimated its cover by the respective taxa (in %). To examine whether the two *Lamium* subsp. differ in performance, we randomly selected three plant individuals from each patch. As measures of performance, we recorded the numbers of stolons and inflorescences, the number of nodes per stolon, and measured the length of the stolons for each plant (in cm).

### 4.3. Growth of L. g. argentatum and L. g. galeobdolon Patches over 7 Years

To examine whether patches of the two *Lamium* subspecies differ in growth, we selected six patches of similar size (0.8–1.2 m^2^) from both taxa in the Allschwil forest near Basel. We marked the 12 patches and measured their area to the nearest 0.1 m^2^ in September 2010. We repeated the measurement for each patch every September in the years 2011–2017.

### 4.4. Performance under Standardized Conditions

“Pot experiments” and “common garden experiments”, as complements to field measurements, allow the investigation of plants under standardized conditions without distracting effects of heterogeneous environmental factors. We conducted a pot experiment to investigate whether small-scale differences in environmental conditions at the sites of origin (e.g., soil properties) could explain differences in patch growth (a measure of performance) between the two taxa. We randomly sampled 10 stolons with each five nodes from both subspecies in the 12 forests (hereafter origin) described in Section 4.1 (Table 1) in September 2016. From each stolon we made fragments of the five nodes with primordial roots and two leaves and a proximal and distal internode length of 1.5 cm. We planted 1200 stolon fragments in pots (diameter: 24 cm; height: 24 cm) filled with standard garden soil (Belflor, Ricoter, Aarberg, Switzerland) in October 2016. In each pot, we planted the five fragments of the same stolon 1 cm below the soil layer, resulting in a total of 240 pots. The pots were placed on a flat roof in the city of Basel. To avoid differences in light exposure, the positions of the pots were changed each week. To assess the establishment of juvenile plants, we checked the pots weekly for leaf sprouting and stolon shoots. At the end of the experiment in July 2017, we harvested the juvenile plants and determined their above-ground and below-ground biomass by drying the plants at 80 °C for 48 h.

### 4.5. Impact on Native Ground Vegetation and Soil Properties

#### 4.5.1. Native Ground Vegetation

To assess the impact of *L. g. argentatum* on both native forest vegetation and soil properties, we selected 18 sites, each containing patches of the garden-escaped horticultural plant and nearby-situated patches of native *L. g. galeobdolon.* The study sites were evenly distributed among three deciduous forests in the suburbs of Basel, Switzerland. The vegetation type of these forests belongs to the *Galio oderati-Fagetum* association [84]. The most abundant tree species are European beech (*Fagus sylvatica*) and sycamore (*Acer pseudoplatanus*). The ground vegetation is rich in vernal geophytes, including *Ranunculus ficaria*, *Anemone nemorosa* and *Alium ursinum*. The distances between study sites within a forest ranged from 50 to 1280 m and the three forests were 1.7–7.0 km apart.

We installed three adjoining 1-m^2^ plots in the center of one similar-sized patch of both taxa. At a distance of 2–3 m from each patch, we set-up three plots of the same size and same spatial arrangement, but without either *Lamium* subspecies (hereafter referred to as control plots). This small-scale arrangement of three types of plots minimized potential differences in site characteristics.

In a plot chosen at random from the three plots in the patches and from the three control plots, we determined all plants in the ground vegetation (herbs and woody plants up to a height of 40 cm) to the species and visually estimated their cover using the Domin scale [85]. To complete the species list of this patch, we carefully searched for additional species in the other two plots. Thus, the abundance of plant species was based on one plot of 1 m^2^, while for the records of plant species richness, three plots (3 m^2^) were considered. Plant species were identified and classified as native or non-native to Switzerland according to Lauber et al. [86]. Plant surveys were carried out between April and September 2019, once in spring and once in fall.

#### 4.5.2. Physical and Chemical Soil Properties

We collected five soil samples in one randomly selected plot from the three plots in all patches and the corresponding control plots using a soil corer (depth 5 cm, diameter 5.05 cm, volume 100 cm^3^) in both late April and early October 2019. The five soil samples from each patch were pooled and sieved (mesh size 2 mm) for further analyses. The same procedure was used for the five soil samples from the control plots. A subsample from each patch and control plot was stored at –80 °C for the assessments of enzyme activity (see Section 4.5.3), soil fungal and bacterial community profile (T-RFLP; see Section 4.5.4), and physiological profiles of soil bacteria (Ecoplates; see Section 4.5.5). We dried the other part of each soil sample at 50 °C for 48 h. We determined soil moisture (%) using the fresh weight to dry weight ratio. Soil pH was assessed in distilled water (1:2.5 soil:water [87]. Total soil organic matter content (SOM, %) was determined as loss-on-ignition of oven-dried soil at 750 °C for 16 h [87]. We assessed plant-available phosphorus by extracting 2 g of soil with 50 mL of acetic acid-sodium acetate buffer (pH 4.8) and shaking for 16 h. After centrifugation at 4000 rpm for 15 min, we determined the phosphorus content (μg PO4^3−^/g) of an aliquot using the molybdenum blue method [88].

#### 4.5.3. Soil Enzyme Activity

We determined the activities of β-1,4-glucosidase (EC 3.2.1.21) and acid phosphatase (EC 3.1.3.2) of soil from each 18 patches of the two taxa and the corresponding control plots following the protocol of Saiya-Cork et al. [89]. Details of the method are presented in Appendix B.

#### 4.5.4. Soil Fungal and Bacterial Community Profiles

We extracted DNA of fungi and bacteria in soil samples from each of the 18 patches of both taxa and the corresponding control plots to examine whether *L. g. argentatum* changes soil fungal and bacterial diversity. We used the T-RFLP method (terminal restriction fragment length polymorphism [90]) to examine potential differences in soil fungal and bacterial community profiles. Microbial diversity activity was expressed as average well color development (AWCD) corrected by the diffusion of the wells (absorbance at 750 nm) and the color development in the control well. Substrate richness was calculated as the number of different substrates used following Zak et al. [91]. The procedure is presented in detail in Appendix B.

#### 4.5.5. Physiological Profiles of Soil Bacteria

Ecoplates assess physiological profiles of bacterial communities. We evaluated the potential influence of *L. g. argentatum* on the activity and metabolic diversity of soil bacterial communities using Biolog Ecoplates^TM^ (Biolog Inc., Hayward, CA, USA). We examined soil bacterial communities from 18 *L. g. argentatum* patches, 18 nearby *L. galedobdolon* patches, and from each of the 18 control plots of both taxa, resulting in a total of 72 samples. Details of the procedure are presented in Appendix B.

### 4.6. Statistical Analyses

All statistical analyses were performed in R [92]. We applied a Contingency test to assess whether the frequencies of *L. g. argentatum* occurrence (expressed in four classes) in the 93 mapping areas differed between the two surveys (1980–1996 vs. 2010–2020). We used general linear models (GLM) to determine potential differences in performance measures (mean values of the following response variables: patch size, plant cover, number of inflorescences, number of stolons, stolon length per plant, number of nodes; in all variables *n* = 12) between the two *Lamium* subspecies. Subspecies and site (*n* = 12 forests) were included in the GLM models with gamma-distributed errors considering the distribution of the data.

To avoid temporal pseudo-replication, we used linear mixed models (LME) in the *nlme* package [93] to analyze the effects of the two *Lamium* subspecies, origin of the stolon fragments (*n* = 12 forests), and elapsed time on the response variable percentage of juvenile plants that emerged from stolon fragments per month (arcsine square-root transformed; *n* = 720). Subspecies and origin were included as fixed factor and elapsed time as random factor. We used analyses of variance (ANOVA) to assess the effects of the two *Lamium* subspecies, origin (12 forests), and node position in the original stolon (four positions) on the response variable percentage of stolon emerged (arcsine square root-transformed; *n* = 120), and on the response variable total biomass of the emerged juvenile plants (*n* = 120).

We used linear mixed models (LME) to analyze the effects of the two *Lamium* subspecies, forest (three forests), and season (spring, autumn) on the response variables above-ground plant species richness (number of plant species/3 m^2^) and soil properties. Subspecies, forest and season were included in the model as fixed factors, and plot nested in season as a random factor. In all analyses, we considered the difference in a variable between plots containing one of the *Lamium* subspecies and their corresponding control plots (*n* = 72). Preliminary analyses revealed inter-correlations between several chemical and physical characteristics of the soil (soil pH vs. SOM: *r_s_* = 0.50, *n* = 144, *p* < 0.001; SOM vs. plant-available phosphorus: *r_s_* = 0.26, *n* = 144, *p* < 0.001). In addition, β-glucosidase activity and acid phosphatase activity correlated (*r_s_* = 0.60, *n* = 144, *p* < 0.001) as did the number of bacterial and fungal OTUs (*r_s_* = 0.24, *n* = 144, *p* = 0.004). In the final analysis, we therefore considered soil moisture, plant-available phosphorus, acid phosphatase activity, numbers of bacterial OTUs, AWCD (average well color development, 168 h) and substrate richness (168 h).

We used analysis of similarity (ANOSIM) in the *vegan* package [94] to examine differences in plant species composition and bacterial CLPP patterns between plot pairs (plots containing either of the two subspecies and their corresponding control plots; in both analyses: *n* = 144). ANOSIM is a nonparametric permutation procedure that allows comparison of between-group and within-group dissimilarities [95]. The procedure calculates *R* statistics ranging from −1 to 1. *R* = 0 indicates completely random grouping, while *R* = 1 when all replicates within groups are more similar than all replicates between groups.

## 5. Conclusions

Our study showed that the presence of *Lamium g. argentatum* reduced native plant species richness, changed the species composition and altered several soil properties in suburban deciduous forests. Our results demonstrate that *L. g. argentatum* should be considered an invasive taxon like *Impatiens glandulifera* [15] and *Prunus laurocerasus* [30]. We suggest that *L. g. argentatum* should be added to the Black List of Invasive Species in Switzerland as well as in other countries. The occurrence of *L. g. argentatum* can often be associated with the presence of nearby garden waste dumping sites. Dumping of garden waste should be avoided in forests. Prevention of invasion of non-native plants can further be improved by raising awareness of both garden owners and the general public about negative effects of non-native species that have become invasive [96]. McKinney [97] stressed the need to develop an ecologically better-informed public, as very often people cannot tell whether a species is native or not.

## Figures and Tables

**Figure 1 plants-12-01527-f001:**
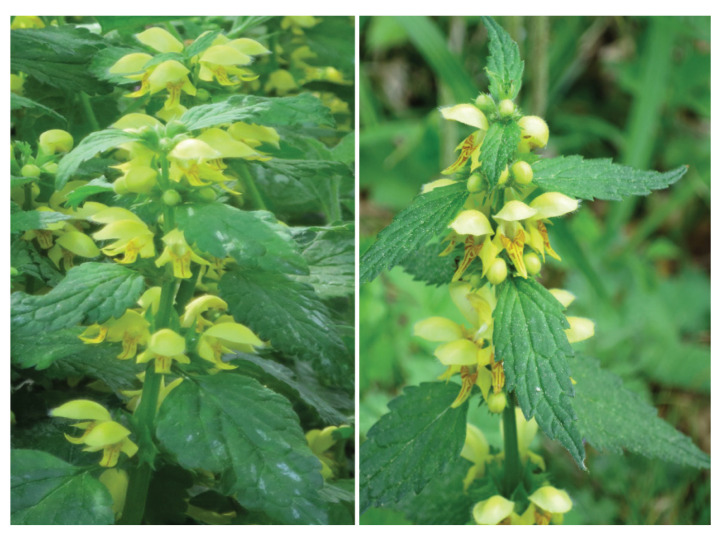
Inflorescences of the horticultural *Lamium galeobdolon* subsp. *argentatum* (**left**) and the native *Lamium galeobdolon* subsp. *galeobdolon* (**right**) in the same forest. Photos: H.-P. Rusterholz.

**Figure 2 plants-12-01527-f002:**
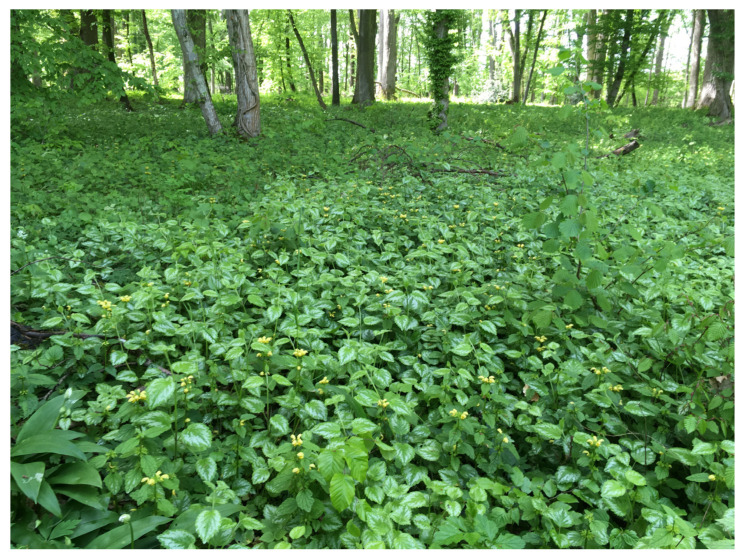
Patch of *Lamium galeobdolon* subsp. *argentatum* in the forest of Allschwil near Basel, Switzerland. Photo: K. Huber.

**Figure 3 plants-12-01527-f003:**
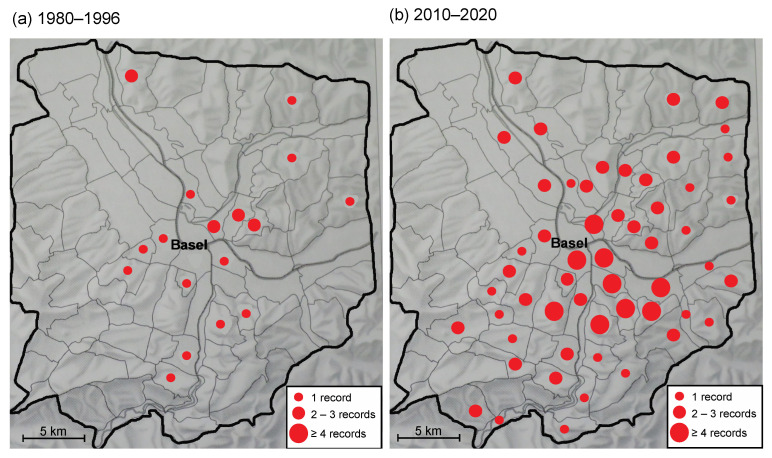
Frequency of *Lamium galeobdolon* subsp. *argentatum* records in 92 mapping areas in the region of Basel (Switzerland) in the years 1980–1996 (**a**), and in 2010–2020 (**b**). Data 1980–1996 from Brodtbeck et al. [32].

**Figure 4 plants-12-01527-f004:**
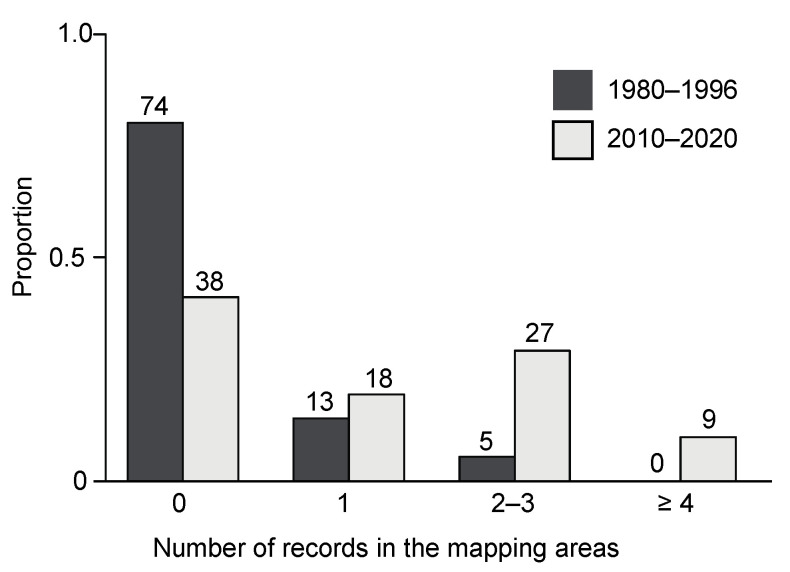
Frequency distribution of *Lamium galeobdolon* subsp. *argentatum* records in 92 mapping areas in the region of Basel (Switzerland) in the years 1980–1996, and in 2010–2020.

**Figure 5 plants-12-01527-f005:**
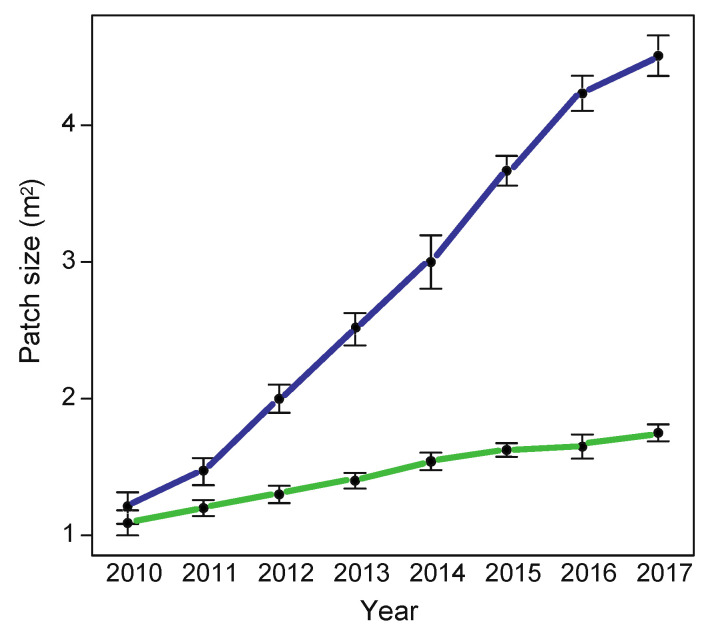
Size of *Lamium galeobdolon* subsp. *argentatum* (blue line) and *Lamium galeobdolon* subsp. *galeobdolon* (green line) patches (in m^2^) in 2010–2017. Mean values ± s.e. are shown; *n* = 6 for both taxa.

**Figure 6 plants-12-01527-f006:**
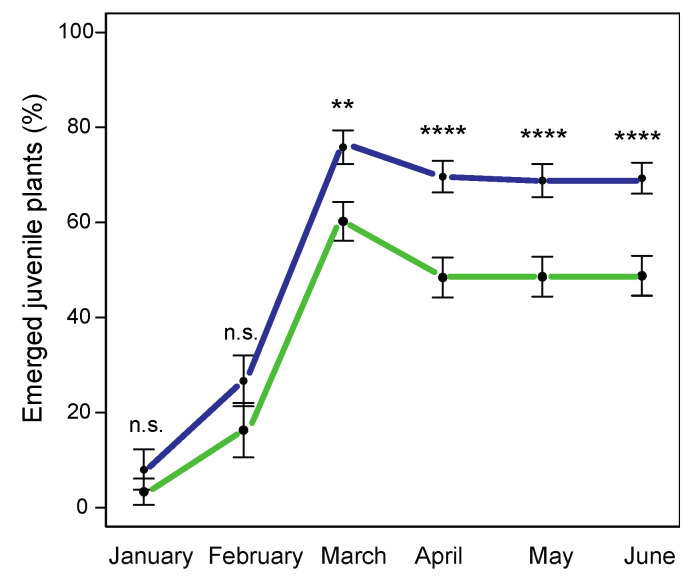
Accumulation curves of emerged juvenile plants of *Lamium galeobdolon* subsp. *argentatum* (blue line; in %) and *Lamium galeobdolon* subsp. *galeobdolon* (green line) from January to June. Asterisks indicate significant differences between the two subspecies in a particular month (** *p* < 0.01, **** *p* < 0.0001, n.s. not significant). Differences were tested using *Chi*^2^ tests. Mean values of all populations ± s.e. are shown; *n* = 12.

**Figure 7 plants-12-01527-f007:**
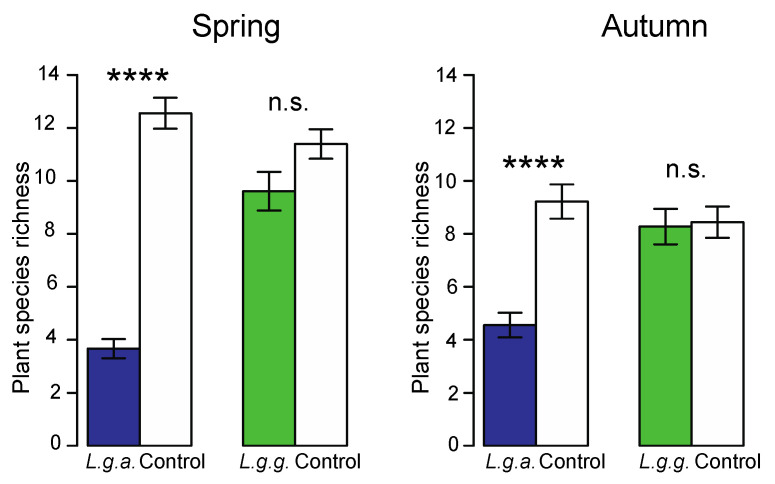
Plant species richness (number of species per 3 m^2^) recorded in *Lamium galeobdolon* subsp. *argentatum* and *Lamium galeobdolon* subsp. *galeobdolon* plots, and the corresponding control plots in both spring and autumn. Mean values ± s.e. are shown, *n* = 18. Differences were tested using paired *t*-tests; **** *p* < 0.0001, n.s. not significant).

**Figure 8 plants-12-01527-f008:**
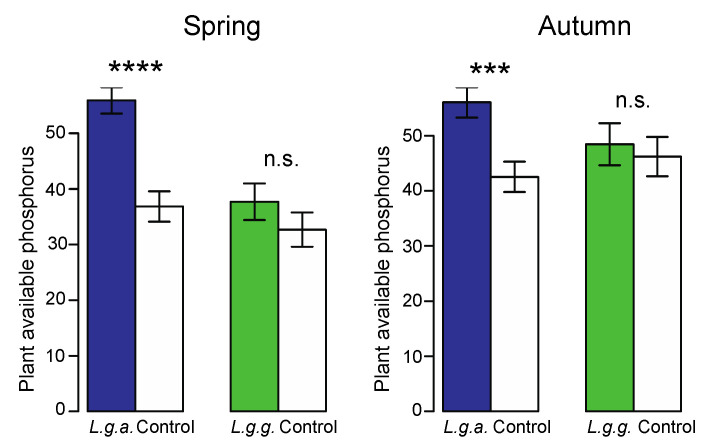
Plant-available phosphorus in *Lamium galeobdolon* subsp. *argentatum* plots, *Lamium galeobdolon* subsp. *galeobdolon* plots and corresponding control plots in both spring and autumn. Mean values ± s.e. are shown, *n* = 18. Differences were tested using paired *t*-tests; *** *p* < 0.001, **** *p* < 0.0001, n.s. not significant).

**Figure 9 plants-12-01527-f009:**
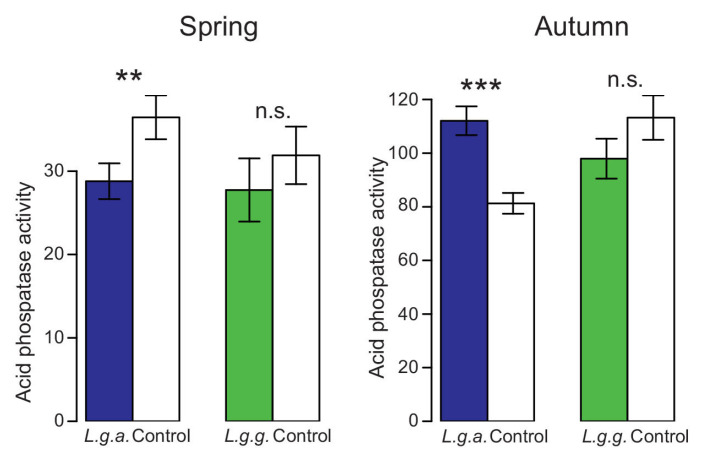
Acid phosphatase activity (nmol MUB released per g soil and h) in *Lamium galeobdolon* subsp. *argentatum* plots, *Lamium galeobdolon* subsp. *galeobdolon* plots and corresponding control plots in both spring and autumn. Mean values ± s.e. are shown, *n* = 18. Differences were tested using paired *t*-tests; ** *p* < 0.01, *** *p* < 0.001, n.s. not significant).

**Figure 10 plants-12-01527-f010:**
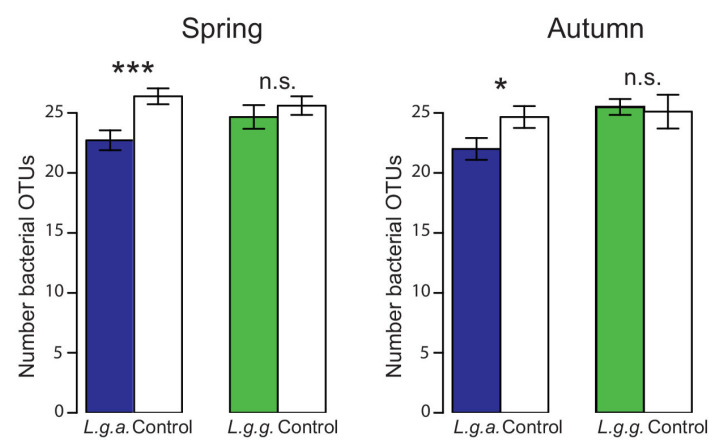
Number of bacterial OTUs in *Lamium galeobdolon* subsp. *argentatum* plots, *Lamium galeobdolon* subsp. *galeobdolon* plots and corresponding control plots in both spring and autumn. Mean values ± s.e. are shown, *n* = 18. Differences were tested using paired *t*-tests; * *p* < 0.05, *** *p* < 0.001, n.s. not significant).

**Figure 11 plants-12-01527-f011:**
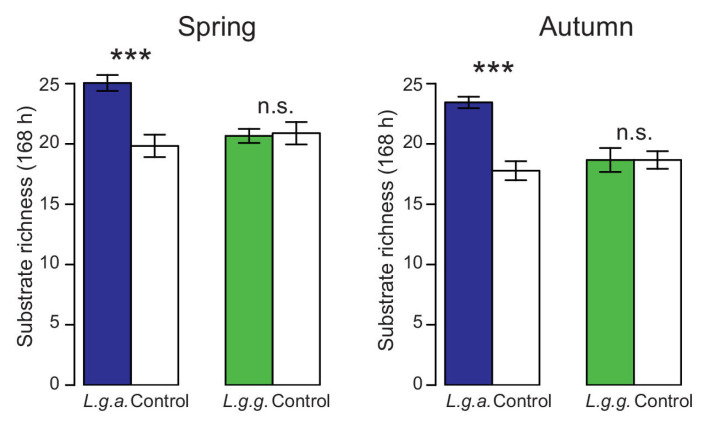
Substrate richness (168 h) in *Lamium galeobdolon* subsp. *argentatum* plots, *Lamium galeobdolon* subsp. *galeobdolon* plots and corresponding control plots in both spring and autumn. Mean values ± s.e. are shown, *n* = 18. Differences were tested using paired *t*-tests; *** *p* < 0.001, n.s. not significant).

**Table 1 plants-12-01527-t001:** Plant traits of horticultural *Lamium galeobdolon* subsp. *argentatum* and native *Lamium galeobdolon* subsp. *galeobdolon* in 12 forests.

Site	Coordinates N/E	Subspecies	Patch Size (m^2^) ^1^	Plant Cover (%) ^1^	Number of Inflorescences ^2^	Number of Stolons ^2^	Stolon Length (cm) ^2^	Number of Nodes ^2^
Allschwil	47.53119/7.54510	Horticultural	57.5 (7.5–450.0)	77.5 (60–90)	0.5 (0–4)	3.5 (2–5)	79.5 (58.0–117.0)	9 (5–14)
		Native	4.5 (1.0–10.0)	67.5 (50–80)	0.5 (0–2)	3 (1–5)	57.5 (40.0–111.0)	6.5 (3–12)
Bottmingen	47.50928/7.56916	Horticultural	11.0 (5.0–250.0)	77.5 (60–80)	1.5 (0–4)	3.5 (1–5)	56.5 (41.0–107.0)	7 (5–11)
		Native	3.5 (2.0–7.5)	50.0 (50–75)	1.5 (0–3)	3 (1–5)	60.0 (35.0–112.0)	6.5 (3–12)
Riehen	47.57129/7.65276	Horticultural	100.0 (15.0–500.0)	90.0 (50–100)	1.0 (0–2)	3 (3–5)	86.5 (47.0–125.0)	7 (5–11)
		Native	5.0 (2.0–10.0)	70.5 (50–80)	1.0 (0–2)	3 (2–3)	50.0 (42.0–97.0)	5.5 (4–7)
Zell (Ger)	47.69662/7.85479	Horticultural	12.5 (5.0–650.0)	75 (60–80)	0.0 (0–3)	4 (1–6)	59.0 (40.0–101.0)	5 (5–7)
		Native	5.0 (3.0–10.0)	68 (50–90)	1.0 (0–3)	2 (1–3)	50.5 (47.0–69.0)	6 (4–9)
Seprais	47.36978/7.22973	Horticultural	10.5 (5.0–12.0)	63 (40–75)	0.0 (0–0)	3 (2–5)	70.0 (52.0–91.0)	9 (5–11)
		Native	2.0 (1.0–4.0)	75 (50–80)	1.0 (0–2)	3 (1–4)	52.0 (35.0–61.0)	4.5 (3–6)
Allaman	46.46913/6.41164	Horticultural	155.0 (18.0–550.0)	55 (40–70)	1.0 (0–2)	4 (2–5)	72.5 (48.0–101.0)	4 (3–6)
		Native	3.5 (1.0 –10.0)	90 (50–90)	1.0 (0–2)	2 (1–4)	62.0 (21.0–107.0)	6 (3–8)
Köniz	46.93223/7.41164	Horticultural	10.0 (2.0–25.0)	75 (50–100)	2.0 (0–4)	3 (3–5)	91.5 (50.0–101.0)	6 (4–9)
		Native	5.5 (2.0–9.0)	65 (10–75)	2.0 (0–3)	2 (2–5)	66.5 (41.0–121.0)	5 (4–9)
Oberwangen	46.92649/7.46436	Horticultural	10.0 (4.0–12.0)	75 (50–80)	1.0 (0–2)	3 (2–5)	61.0 (41.0–101.0)	9.5 (7–11)
		Native	5.0 (4.0–12.0)	75 (10–80)	1.0 (0–2)	3 (1–4)	66.5 (41.0–141.0)	6.5 (4–11)
Wabern	46.92649/7.46436	Horticultural	9.0 (6.0–25.0)	77.5 (60–90)	0.0 (0–2)	3 (3–5)	90.5 (51.0–123.0)	6 (4–9)
		Native	4.0 (2.0–8.0)	70.5 (40–90)	0.5 (0–2)	3 (1–4)	57.5 (43.0–96.0)	6 (3–11)
Hägglingen	47.37990/8.26482	Horticultural	57.5 (15.0–325.0)	82.5 (75–90)	1.0 (0–2)	3 (2–4)	51.5 (34.0–118.0)	5 (3–13)
		Native	13.5 (6.0–100.0)	72.5 (60–90)	1.0 (0–2)	2 (1–3)	56.0 (45.0–112.0)	6 (4–11)
Zürich	47.38537/8.57158	Horticultural	11.0 (6.0–22.5)	75 (60–90)	1.0 (0–2)	4 (2–5)	63.5 (41.0–109.0)	7 (4–11)
		Native	5.5 (2.0–9.0)	90 (60–100)	0.5 (0–4)	3 (2–4)	56.0 (28.0–128.0)	7 (5–11)
Zollikon	47.34246/8.58590	Horticultural	11.0 (6.0–36.0)	77.5 (60–90)	0.0 (0–0)	3 (2–5)	70.0 (47.0–107.0)	7 (5–8)
		Native	10.0 (4.0–30.0)	75 (60–90)	1.0 (0–4)	2.5 (1–5)	70.5 (58.0–105.0)	6 (3–10)

^1^ *n* = 6 per subspecies and site; ^2^ *n* = 18 per subspecies and site; mean values and ranges (in parentheses) are indicated.

**Table 2 plants-12-01527-t002:** Summary of linear mixed effects models testing the effects of subspecies (*L. g. argentatum* vs. *L. g. galedobdolon*), forests (three sites), season (spring, autumn) on plant species richness (number of species per 3 m^2^) and various soil properties. In all analyses, we considered the difference in a variable between plots containing one of the *Lamium* subspecies and their corresponding control plots.

	Plant Species Richness	Plant-Available Phosphorus ^1^	Soil Moisture	Acid Phosphatase Activity ^2^	Number of Bacterial OTUs	AWCD (168 h)	Substrate Richness (168 h)
Subspecies	F_1,30_ = 76.29, ***p* < 0.001**	F_1,30_ = 8.16, ***p* = 0.006**	F_1,30_ = 0.66, *p* = 0.422	F_1,30_ = 12.86, ***p* = 0.001**	F_1,30_ = 7.76, ***p* = 0.001**	F_1,30_ = 2.76, *p* = 0.107	F_1,30_ = 22.72, ***p* < 0.001**
Forest	F_2,30_ = 0.49, *p* = 0.619	F_2,30_ = 0.99, *p* = 0.384	F_2,30_ = 1.25, *p* = 0.300	F_2,30_ = 2.25, *p* = 0.123	F_2,30_ = 1.31, *p* = 0.285	F_2,30_ = 0.78, *p* = 0.467	F_2,30_ = 1.80, *p* = 0.183
Season	F_1,30_ = 34.91, ***p* < 0.001**	F_1,30_ = 3.95, *p* = 0.056	F_1,30_ = 1.95, *p* = 0.173	F_1,30_ = 15.52, ***p* = 0.001**	F_1,30_ =1.35, *p* = 0.254	F_1,30_ = 0.77, *p* = 0.386	F_1,30_ = 0.12, *p* = 0.729
Subspecies × Forest	F_2,30_ = 0.84, *p* = 0.442	F_2,30_ = 0.34, *p* = 0.716	F_2,30_ = 1.05, *p* = 0.365	F_2,30_ = 1.81, *p* = 0.182	F_2,30_ = 4.64, ***p* = 0.018**	F_2,30_ = 0.46, *p* = 0.638	F_2,30_ = 0.74, *p* = 0.486
Subspecies × Season	F_1,30_ = 6.99, ***p* = 0.013**	F_1,30_ = 0.14, *p* = 0.714	F_1,30_ = 4.85, ***p* = 0.035**	F_1,30_ = 31.42, ***p* < 0.001**	F_1,30_ = 0.03, *p* = 0.869	F_1,30_ = 0.65, *p* = 0.421	F_1,30_ = 0.01, *p* = 0.908
Subspecies × Forest × Season	F_4,30_ = 1.60, *p* = 0.199	F_4,30_ = 1.73, *p* = 0.170	F_4,30_ = 0.42, *p* = 0.796	F_4,30_ = 2.57, *p* = 0.058	F_4,30_ = 5.48, ***p* = 0.002**	F_4,30_ = 1.37, *p* = 0.269	F_4,30_ = 2.08, *p* = 0.109

Significant *p*-values (< 0.05) are indicated in bold. ^1^ mg PO_4_^3-^ per g soil. ^2^ nmol MUB released per g soil and h.

## Data Availability

Data presented in this study are available in the article and its Supporting Information Files (Appendix A).

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
