# Peer review of "Invasion of a Horticultural Plant into Forests: Lamium galeobdolon argentatum Affects Native Above-Ground Vegetation and Soil Properties"

_plants, 2023, doi:10.3390/plants12071527_

Round 1

Reviewer 1 Report

General comments

The objectives of the study are to stress the particularities of Lamium galeobdolon var. argentatum to establish and understand its invasiveness. The authors have used a variety of methodologies (historical data analysis, field observations and samplings, field and garden experiments) to reach their objectives. All the text and presentation are neat. The results and their discussion support their hypotheses.

Particular comments

- L94 I suggest to change all through the text ‘Common garden experiment’ by ‘Performances under controlled conditions’

- Fig. 7 to 11: give p-levels on the plots

- L292 In this section and in the next one, only L. g. argentatum is discussed. What about the spread and the performance in the wild of other invasive species?

-L341 ‘change ‘resouces’ to ‘resources’

- L429 Give how the changes were tested (chi-squared test)

- L545 Justify why you used gamma-distributed error (with log link?)

- L 549 According to my understanding of the datasets (% of juvenile plants at nodes, i.e. you looked if there is a plant at each node and then compute the proportion of nodes with a plant?), I think you better have to use logit models rather than LMM with transformed data.

- L578 I suggest to add some examples of invasive species having the same signature as L. g. argentatum

Author Response

We considered the remarks of the reviewers and made the following improvements (our reply in italics).

Reviewer 1

General comments

The objectives of the study are to stress the particularities of Lamium galeobdolon var. argentatum to establish and understand its invasiveness. The authors have used a variety of methodologies (historical data analysis, field observations and samplings, field and garden experiments) to reach their objectives. All the text and presentation are neat. The results and their discussion support their hypotheses.

Particular comments

L94 I suggest to change all through the text ‘Common garden experiment’ by ‘Performances under controlled conditions’

We followed this suggestion when it was possible. However, we did not conduct the experiment under controlled conditions. “Standardized conditions” is a more appropriate term. We improved the text.

- Fig. 7 to 11: give p-levels on the plots

We inserted the significance levels in the Figs 7–11 and extended the legends to these figures.

- L292 In this section and in the next one, only L. g. argentatum is discussed. What about the spread and the performance in the wild of other invasive species?

The spread of other invasive plants and their impact has already been mentioned in the text. We improved this paragraph by adding two examples.

-L341 ‘change ‘resouces’ to ‘resources’

Done

- L429 Give how the changes were tested (chi-squared test)

Please note that this information is already given in Section 4.6 Statistical analyses, first paragraph.

- L545 Justify why you used gamma-distributed error (with log link?)

We considered the distribution of the data. In this case gamma-distributed error is the most adequate solution (confirmed by a statistician).

- L 549 According to my understanding of the datasets (% of juvenile plants at nodes, i.e. you looked if there is a plant at each node and then compute the proportion of nodes with a plant?), I think you better have to use logit models rather than LMM with transformed data.

The reviewer is correct when the proportion of nodes were considered. However, we considered percentage values in our analyses and therefore the models and transformation used are adequate (confirmed by a statistician). Both types of analyses revealed the same results.

L578 I suggest to add some examples of invasive species having the same signature as L. g. argentatum

Following the suggestion of the reviewer we inserted two examples.

Reviewer 2 Report

This study examines the traits of native Lamium galeobdolon galeobdolon and invasive L. g. argentatum to assess characteristics that explain the invasiveness of the introduced species. A large number of plant, soil and microbial characteristics are sufficiently different to understand the causes of invasion as well as effects on soil characteristics and native plant richness. The authors conclude with advice for management to reduce future new introductions. The study is unique in comparing two closely related taxa, where the invasive taxon has not been widely recognized as a non-native species.

The manuscript is well written, and the Discussion is interesting and clearly explains the Results.  However, a number of points, especially in the Results, were unclear but could be addressed in a revision, and I noted several minor edits:

25 “considered as invasive taxon.”

CHANGE TO

 considered an invasive taxon.

58 “Lamiacea”

spell as Lamiaceae

66 Does Lamium galeobdolon montanum also occur in the study area?

77 ”L. g. argentatum has no natural distribution area”

What does this mean? It is not known to be native to any land area, so its origin is unknown?

Results

Fig. 6 legend—explain in the legend that these are the means of all populations at all sites [correct?].

The paragraph beginning l. 186 has some confusing points. Do these data also refer to Fig. 6? 186: “Stolons only appeared between June and July”  but July does not appear in Fig. 6.

187: “The two subspecies did not differ in stolon emergence” but Fig. 6 shows significant differences in juvenile plant emergence. Do juvenile plants arise from stolon nodes? Please clarify these points.

197 I recommend adding a figure or table to show biomass results. This can go into an appendix, if the authors feel they have too many figures.

209 “….45 (73.8%) in the control plots.” Should there be two values of total richness for control plots associated with L.g.a. and L.g.g.? There were separate control plots for L.g.a. and L.g.g., as shown in Fig. 7 that reports richness/unit area.

341 “resouces” CHANGE TO resources

352 …“the effect of clonal fragment regeneration on invasiveness of non-native plants remains controversial.“

A better way to say this may be:

….the effect of clonal fragment regeneration on invasiveness of non-native plants is variable and may depend on species.

379 “Ambrosia artemisiafolia” CHANGE SPELLING TO  artemisiifolia

398 “….magnitude of cover the invasive….”

CHANGE TO

magnitude of cover of the invasive….

400 “soil microbial diversity was less diverse”

CHANGE TO

soil microbial diversity was lower

400 “Thus, the reduced bacterial richness recorded in our study could be due to the reduced plant species richness recorded in L. g. argentatum plots. However, this was not the case in our study.”

I don’t understand. Since both bacterial OTUs and plant richness were reduced in L.g.a. plots, why do you conclude with “this was not the case in our study.” Please clarify.

410 “We also recorded an increase in substrate richness in soils invaded by L. g. argentatum. This result is supported by various other studies….”

Discussion on elevated soil P in L.g.a. plots could also be expanded. For instance, L.g.a. might be colonizing sites that are initially higher in soil P. Alternatively, P mineralization may increase in the L.g.a. rhizosphere, perhaps related to higher acid phosphatase activity, or higher substrate richness use activity?

Author Response

We considered the remarks of the reviewers and made the following improvements (our reply in italics).

Reviewer 2

This study examines the traits of native Lamium galeobdolon galeobdolon and invasive L. g. argentatum to assess characteristics that explain the invasiveness of the introduced species. A large number of plant, soil and microbial characteristics are sufficiently different to understand the causes of invasion as well as effects on soil characteristics and native plant richness. The authors conclude with advice for management to reduce future new introductions. The study is unique in comparing two closely related taxa, where the invasive taxon has not been widely recognized as a non-native species.

The manuscript is well written, and the Discussion is interesting and clearly explains the Results. However, a number of points, especially in the Results, were unclear but could be addressed in a revision, and I noted several minor edits:

 25 “considered as invasive taxon.” CHANGE TO considered an invasive taxon.

Done

 58 “Lamiacea” spell as Lamiaceae

Done

66 Does Lamium galeobdolon montanum also occur in the study area?

Lamium galeobdolon montanum occurs at low abundance in the study region. We inserted this information in the text.

77 ”L. g. argentatum has no natural distribution area” What does this mean? It is not known to be native to any land area, so its origin is unknown? 

As a horticultural breed this taxon has no natural distribution area. We improved the text.

Results

Fig. 6 legend—explain in the legend that these are the means of all populations at all sites [correct?].

Yes, means of all populations are presented (n = 12). We improved the legend.

 The paragraph beginning l. 186 has some confusing points. Do these data also refer to Fig. 6? 186: “Stolons only appeared between June and July” but July does not appear in Fig. 6.

This is a misunderstanding. Our former text was not clear enough. This paragraph relates to stolons emerging from juvenile plants (not to Fig. 6). We improved the text to clarify this point.

187: “The two subspecies did not differ in stolon emergence” but Fig. 6 shows significant differences in juvenile plant emergence. Do juvenile plants arise from stolon nodes? Please clarify these points.

We improved the text to clarify this point.

197 I recommend adding a figure or table to show biomass results. This can go into an appendix, if the authors feel they have too many figures.

Data on biomass are already shown in Table S3. We improved the text and refer now to this table.

209 “….45 (73.8%) in the control plots.” Should there be two values of total richness for control plots associated with L.g.a. and L.g.g.? There were separate control plots for L.g.a. and L.g.g., as shown in Fig. 7 that reports richness/unit area.

Following the advice of the reviewer we present separate data for control plots of L.g.a. and control plots of L.g.g.

 341 “resouces” CHANGE TO resources

Done

352 …“the effect of clonal fragment regeneration on invasiveness of non-native plants remains controversial.“ A better way to say this may be: ….the effect of clonal fragment regeneration on invasiveness of non-native plants is variable and may depend on species.

Done

379 “Ambrosia artemisiafolia” CHANGE SPELLING TO artemisiifolia

Done

 398 “….magnitude of cover the invasive….” CHANGE TO magnitude of cover of the invasive….

Done

 400 “soil microbial diversity was less diverse” CHANGE TO soil microbial diversity was lower

Done

400 “Thus, the reduced bacterial richness recorded in our study could be due to the reduced plant species richness recorded in L. g. argentatum plots. However, this was not the case in our study.” I don’t understand. Since both bacterial OTUs and plant richness were reduced in L.g.a. plots, why do you conclude with “this was not the case in our study.” Please clarify.

We clarified this point by amending the text.

410 “We also recorded an increase in substrate richness in soils invaded by L. g. argentatum. This result is supported by various other studies….” Discussion on elevated soil P in L.g.a. plots could also be expanded. For instance, L.g.a. might be colonizing sites that are initially higher in soil P. Alternatively, P mineralization may increase in the L.g.a. rhizosphere, perhaps related to higher acid phosphatase activity, or higher substrate richness use activity?

Please note that we discussed the plant-available phosphorus in the soil on line 377ff (and not on line 410). Following the suggestions of the reviewer we added three sentences to presnt possible explanations.